# Biportal Endoscopic Radiofrequency Ablation of the Sacroiliac Joint Complex in the Treatment of Chronic Low Back Pain: A Technical Note with 1-Year Follow-Up

**DOI:** 10.3390/diagnostics13020229

**Published:** 2023-01-08

**Authors:** Chun Tseng, Kuo-Tai Chen, Yi-Chin Fong, Chung-Wei Lin, Li-Wei Sun, Chien-Min Chen, Guan-Chyuan Wang

**Affiliations:** 1Graduate Institute of Biomedical Sciences, China Medical University, Taichung 404, Taiwan; 2Department of Orthopedic Surgery, China Medical University Beigang Hospital, Yunlin 651, Taiwan; 3Department of Neurosurgery, Chang Gung Memorial Hospital Chiayi, Chiayi 613, Taiwan; 4Department of Sports Medicine, College of Health Care, China Medical University, Taichung 404, Taiwan; 5Department of Orthopedic Surgery, China Medical University Hospital, Taichung 404, Taiwan; 6Division of Neurosurgery, Department of Surgery, Changhua Christian Hospital, Changhua 500, Taiwan; 7Department of Leisure Industry Management, National Chin-Yi University of Technology, Taichung 411, Taiwan; 8Department of Neurosurgery, Mennonite Christian Hospital Foundation, Hualien 970, Taiwan

**Keywords:** sacroiliac joint, endoscopy, radiofrequency ablation, low back pain

## Abstract

Background: Sacroiliac joint (SIJ) pain is a common source of low back pain. Previously reported management strategies for this pain include conservative treatment, SIJ injection, radiofrequency denervation ablation, and SIJ fusion. Herein, we describe the use of biportal endoscopic radiofrequency ablation (BERA) to treat patients with low back pain. Methods: We included 16 patients who underwent BERA from April 2018 to June 2020. We marked the S1, S2, and S3 foramina and the SIJ line under fluoroscopy. Skin entry points were positioned at 0.5 cm medial to the SIJ line and at the level of the S1 and S2 foramina. Under local anesthesia, we introduced a 30° arthroscope with a 4 mm diameter through the viewing portal; surgical instruments were inserted through another caudal working portal. We ablated the lateral branches of the S1–S3 foramina and L5 dorsal ramus, which were the sources of SIJ pain. Results: Clinically relevant improvements in both visual analog scale and Oswestry Disability Index scores were noted at 1-, 6-, and 12-month follow-up time points after surgery. The overall patient satisfaction score was 89.1%. Conclusions: BERA for SIJ pain treatment has the advantage of directly identifying and ablating the innervating nerve to the joint. Through this technique, an expanded working angle can be obtained compared with traditional single-port endoscopy. Our study demonstrated promising preliminary results.

## 1. Introduction

The sacroiliac joint (SIJ) is a common cause of low back pain, although it can be overlooked as a cause of such pain [1,2]. A history of trauma, inflammatory disease, or spinal surgery are precursors to SIJ pain [3,4,5]. Spinal fusion involving the sacrum or multiple-segment fusion increases the incidence of SIJ pain and can be the source of failed back surgery [3,4,6]. The SIJ is not supposed to have excessive movement and should be stable enough to transfer the body weight to the lower extremities. However, after lumbar spinal fusion, the decrease of lumbar motion might force the SIJ to rotate and increase the stress on the SIJ. It also considered to be a form of adjacent segment degeneration after spinal fusion surgery [1].

To treat SIJ pain, the anatomy of SIJ must be emphasized. The joint space, muscle around the SIJ, ligament supporting the SIJ, and the nerve innervating the SIJ make the SIJ complex. SIJ pain is believed to be caused by the lateral sacral branch that extends from the posterior sacral foramen and innervates the interosseous and dorsal sacroiliac ligaments as well as the joint. A previously reported strategy for SIJ pain management included conservative treatment with stabilization or medications, SIJ injection with local anesthesia or steroids, radiofrequency (RF) denervation ablation, and SIJ fusion [6]. Several studies have shown the longer-lasting efficacy of RF ablation (RFA) of the SIJ complex [7]. The concept constituting the basis for this treatment entails denervating the nociceptive sensory nerve that supplies the SIJ. The nerve innervating the SIJ consists of the L5 dorsal ramus and S1 to S3 lateral branches of the sacral rami and these were the denervation target. Choi et al. treated patients with chronic low back pain secondary to SIJ complex by applying single-port endoscopic RFA to the lateral sacral branches; their results showed promising outcomes with 88.6% patient satisfaction [8,9]. In this paper, we introduce a novel technique that entails the use of biportal endoscopic RFA (BERA) for SIJ pain treatment. We present the surgical procedure and treatment outcome and discuss the potential advantages of our technique. In addition, we depict the surgical steps in a Appendix A.

## 2. Materials and Methods

Our study protocol was reviewed and approved by the China Medical University Hospital (CMUH) Research Ethics Committee (REC; REC Code number: CMUH109-REC2-086).

### 2.1. Patient Selection

We selected the medical records of 16 consecutive patients who had undergone BERA for treating SIJ-related low back pain between April 2018 and June 2020.

We used the following inclusion criteria: having a chief concern of low back pain with signs and symptoms of SIJ involvement on physical examination, undergoing conservative care (involving rest, analgesic administration, and physiotherapy) that failed to alleviate the pain, and having persistent low back pain (despite previous lumbosacral operation or pain procedures) lasting more than 12 weeks. In addition, we stipulated that the included patients had a 50% or higher improvement in pain from baseline according to visual analog scale (VAS) measurements conducted after diagnostic intra-articular and multisite lateral sacral branch blocks of the SIJ complex. Finally, we required the included patients to have undergone 12 months of follow-up.

We also used the following exclusion criteria: having tumors of the SIJ, previously receiving surgery on the SIJ (such as SIJ fusion or posterior plating of the SIJ due to trauma), or having severe comorbid medical conditions.

### 2.2. Clinical Assessment

All patients underwent BERA treatment, and the clinical results were assessed preoperatively and at 1-, 6-, and 12-month time points postoperatively by using an outpatient clinic or phone call questionnaire. VAS and Oswestry Disability Index (ODI) scores were recorded preoperatively and at each follow-up time point. The VAS and ODI were the primary outcome assessments. In addition, a patient satisfaction survey was administered 6 months after surgery. All clinical assessments were performed by a single coresearcher.

### 2.3. Statistical Analysis

Due to smaller patient sample, we used nonparametric statistics. The Wilcoxon signed rank test was used to compare preoperative and postoperative VAS and ODI scores. Statistical significance was set to *p* ≤ 0.05. Statistical analyses were performed and graphs were designed using SPSS version 25 (IBM Corp. (ICC/POK), USA 2017).

#### Surgical Techniques

All patients were placed in the prone position on a radiolucent table. Patients remained awake during the surgery to maintain communication with the surgeon throughout the procedure.

After sterile preparations and draping, an anteroposterior fluoroscopic view was obtained using a C-arm. A transducer was tilted cephalad at 10°–15° to optimally visualize the posterior aspect of the SIJ. We marked the S1–S3 foramina and the SIJ line under fluoroscopy (Figure 1a). Skin entry points for the viewing portal and working portal were positioned at 0.5 cm medial to the SIJ line and at the level of the S1 and S2 foramina, separately. We set the S1 incision as the working portal and the S2 incision as the viewing portal (Figure 1b).

We injected 3 cc of local anesthetic with 1% lidocaine hydrochloride at each entry point and 5 cc of lidocaine to infiltrate into the space between S1 and S2 area. Subsequently, two 0.5 cm skin incisions were made at the entry points. A pair of Kelly forceps was used in each of the incisions, and a blunt supported the space between the erector spinae muscles (multifidus and longissimus muscles) and interosseous ligament overlying the posterior SIJ. After the insertion of the cannula, we introduced a 30° endoscope with a 4 mm diameter (Smith & Nephew, Inc., Watford, England, UK) through the viewing portal (Figure 1c). During the procedure, a saline irrigation pump was connected to the endoscope and set to a pressure of 20–30 mm Hg. Surgical instruments were inserted through the caudal working portal (Figure 1c). After triangulation with the endoscope and control of minor bleeding, the ablation wand was used for debridement of the soft-tissue remnants overlying the muscles and interosseous ligament structures (Appendix A).

We ablated the area between the lateral border of the sacral foramen and the medial border of the SIJ. The lateral branches nerves usually penetrate from the sacral foramen and are accompanied by their nutrition vessels and surrounded by fatty tissue. Therefore, we identified the lateral branches of S1, S2, and S3 in the region lateral to the S1–S3 sacral foramina and ablated them (Figure 2a,b). Subsequently, we tilted the endoscope more cranially to identify the dorsal primary ramus of L5, which is usually located at the cranial-lateral quarter of the S1 foramen [10], occasionally anastomosing to the S1 lateral plexus. The position of the tip of the RF probe could be verified under fluoroscopic guidance if necessary. Throughout the procedure, we provoked SIJ pain by ablating the ligament structure. The patient located the trigger point, which should be consistent with the most uncomfortable point they experienced during their daily activities. We then ablated the ligament structure under the endoscope without violating the foramen structure.

We attempted to visually confirm the lateral branches exiting the sacral foramina and the branches coursing toward the SIJ to ensure accuracy during nerve lesioning (Figure 3). Throughout the procedure, we maintained communication with the patient to assess the pattern and location of pain. We asked the patient if the pattern and location of SIJ pain were associated with each stimulus and to identify which stimulus area caused the most pain. After ablation of the target points, the endoscope and cannula were removed, and the wound was closed with 3-0 nylon simple interrupted sutures (Figure 1d)

## 3. Results

From April 2018 to June 2020, 16 patients underwent BERA at China Medical University, Beigang Hospital. These patients’ data are presented in Table 1. The median preoperative VAS score was 7 (range: 6–8), and the mean preoperative ODI score was 33 (range: 25–48; Figure 4). All patients experienced improvements in VAS and ODI scores at 1 month after surgery, which persisted to 12 months. At the 12-month follow-up, the median VAS score improved to 1.0 (range: 0–3), and the ODI score improved to 10 (range: 5–22; Figure 4). This improvement was statistically significant for both the VAS and ODI scores (*p* < 0.001). The patients’ blood loss was minimal (Table 1). The median duration of the operation from the time of local anesthetic injection to wound closure was 55 (range: 21–65) min. The overall patient satisfaction score was 89.1%, recorded at 6 months after surgery.

## 4. Discussion

Available treatments for intractable SIJ pain range from SIJ injection, RFA, and endoscopic denervation to SIJ fusion. Previous studies have reported prolotherapy involving the injection of hyperosmolar dextrose or platelet-rich plasma into periarticular and intra-articular areas [11,12,13].

RFA approaches, including cooled RFA of the SIJ, have also been reported to be promising approaches with favorable treatment effects [14,15,16]. The goal of RFA is to denervate the dorsal ramus of L5 and the lateral branch of S1–S3, which are thought to be the signal source of pain from the SIJ [17,18]. If the listed treatments have failed, SIJ fusion with a minimally invasive technique might be considered [1,16].

Martin et al. compared the short-term and long-term outcomes of patients with SIJ fusion. Their pooled analysis revealed that on average, the VAS scores decreased from 80.3 to 32.2 and the ODI scores decreased from 56.2 to 34.4 [19]. However, general anesthesia and longer hospital stays were both required for SIJ fusion.

Previous research reported that treatment with RFA for the SIJ complex exhibited longer-lasting efficacy than did other treatment [7]. Compared with SIJ fusion, RFA is a less invasive treatment performed under local anesthesia. Therefore, we considered RFA to be a suitable solution to SIJ pain because it achieved similar pain relief as the aforementioned methods.

Conventional RFA is performed through needle insertion into the area between the dorsal foramina and the SIJ under oblique anterior–posterior X-ray. However, this technique is image guided, and the denervation of the lateral sacral branches is performed without the observation of the structure within the SIJ. Because the lateral branches of S1–S3 travel deep to the long posterior sacroiliac and sacrotuberous ligaments [20], conventional RFA technique tends to ablate the required areas too superficially. A previous study asserted that SIJ pain is generated from both the nerve and the ligament structures, which are barely reachable with conventional techniques [6]. These might contribute to pain recurrence in the long term after conventional RFA [16,21]. Vanaclocha et al. reported pain recurred after 6 months of surgery and regained to pre-RFA status after 72 months [16].

The advantage of using an endoscope in this study was that we could not only identify the lateral sacral branches directly but also manage the pain associated with the attaching ligaments. Many of our patients mentioned that the most uncomfortable area was over the cranial one-third portion of the SIJ, which might be the area where the dorsal ramus of L5 and the lateral branch of S1 converge.

Furthermore, we could stimulate the suspected lateral branch by using the RF probe in a gentle manner to elicit pain and ensure that the correct nerve had been successfully ablated. Two studies on the use of endoscopic RFA in SIJ pain management have already shown promising results. Choi et al. reported a mean VAS score improvement from 6.7 to 2.8 and ODI score improvement from 22.2 to 12.0 at 6 months after surgery [8]. Ibrahim reported a mean VAS score improvement from 7.23 to 2.82 and ODI score improvement from 21.73 to 19.09 at 24 months after surgery [9]. These two studies have used a single-port endoscopic technique; by contrast, we developed BERA to treat SIJ pain and our patients’ clinical outcomes were non-inferior to the previous studies (VAS improvement from 7 to 1 and ODI improvement from 33 to 10).

The BERA technique has several advantages over single-portal endoscopic RFA. First, preparation of a specific single-port endoscope is not required. We used a 4.0 mm elbow endoscope and an ablation wand, both of which are common tools in an orthopedic department (these are the same instruments as arthroscopes). Second, the BERA technique involves less restriction of motion and a wider angle corridor compared with conventional single-port endoscopy (Figure 5). A surgeon can easily see the operative field over the L5 dorsal ramus to the S3 sacral foramina area by switching the viewing and working portals, thus a very steep viewing angle could be overcome (Figure 5). Third, through the proposed BERA technique, a surgeon can ablate the ligament structure and control bleeding more efficiently with an ordinary ablation wand than they could using single-port endoscopic RFA. Our mean duration of operation from the time of local anesthetic injection to wound closure was 18.5 min per side; by contrast, the mean operating time of single-port endoscopic RFA is 26.6 min per side. We have experience with both techniques and consider BERA to be more efficient to create the space available in the operation field.

Our study has several limitations. First, this was a retrospective study with a limited number of cases, and we report only 12-month follow-up data. Some patients might experience pain recurrence in the long run. Second, we did not directly compare the clinical results and perioperative parameters of BERA with those of a single-port surgical technique or other methods. Therefore, we cannot make the conclusion that BERA is superior to other treatment in terms of pain control. In the future, we plan to conduct a meta-analysis to compare the different strategy in treating SIJ pain.

## 5. Conclusions

In this paper, we demonstrate a new technique involving the use of BERA to treat SIJ pain. According to our experience, BERA for SIJ pain treatment has the advantage of directly identifying and ablating the innervating nerve to the joint. It does not require special endoscopic instruments and has a superior working angle to single-port endoscopy. Our patients experienced long-term pain relief and improved physical function with minimal complications.

## Figures and Tables

**Figure 1 diagnostics-13-00229-f001:**
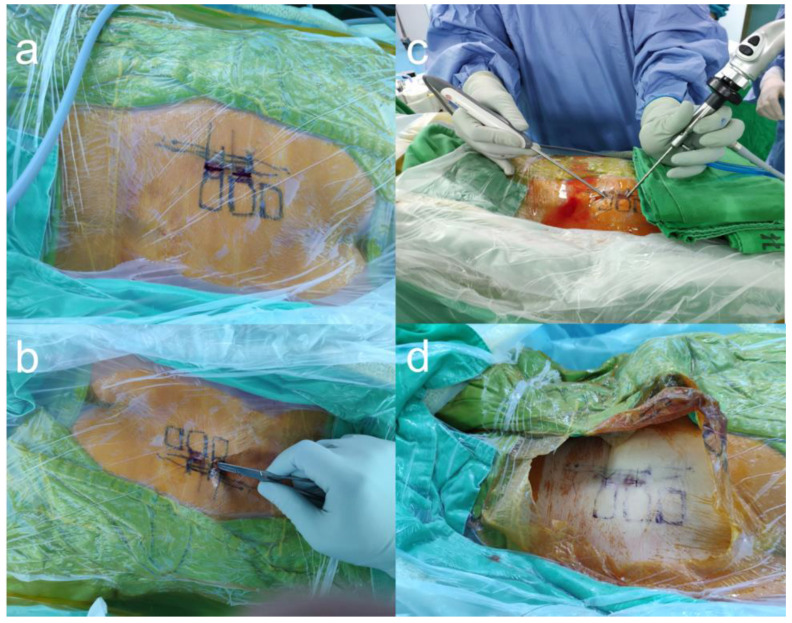
The entry points and working channels of biportal endoscopic radiofrequency ablation. (**a**) Under fluoroscope, the S1–S3 foramina were marked on the skin. (**b**) Two incisions were made, one near the S1 foramen and another near the S2 foramen. (**c**) One foramen was for the endoscopic channel, and the other was for the insertion of the ablation wand. (**d**) Sutured wound after the surgery.

**Figure 2 diagnostics-13-00229-f002:**
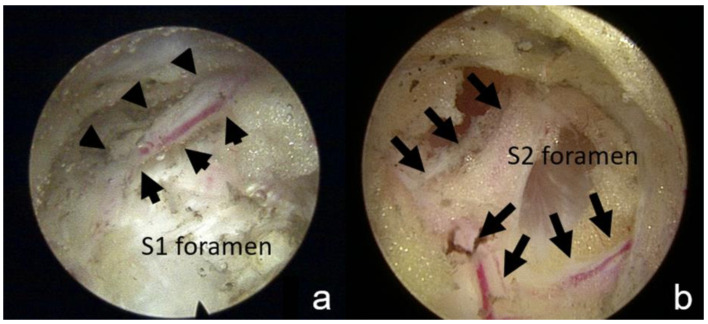
Foramen and sensory nerve under endoscopic view. (**a**) This endoscopic view demonstrates the S2 foramen and the exiting sensory nerve (arrow). (**b**) The sensory nerve is usually accompanied by blood vessels.

**Figure 3 diagnostics-13-00229-f003:**
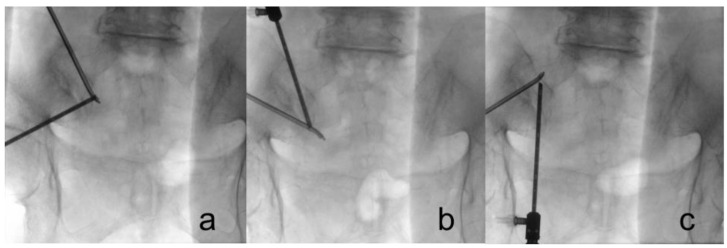
Fluoroscopes were used to guide the ablation point. (**a**) S1 foramen; (**b**) S2 foramen; (**c**) L5 dorsal ramus, which is located at the junction of the sacral alar and promontory.

**Figure 4 diagnostics-13-00229-f004:**
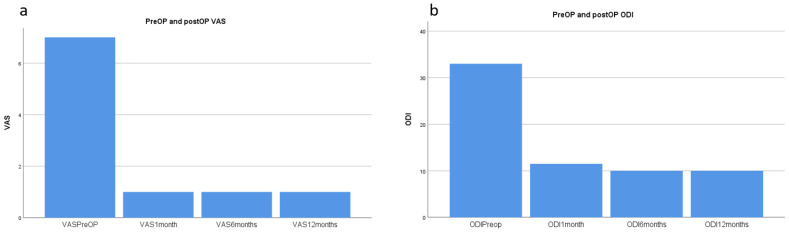
VAS and ODI scores before and after the surgery (**a**) Median VAS score improved from 7 (6–8) to 1 (0–3) after surgery. (**b**) Median ODI score improved from 33 (25–48) to 10 (5–22) after surgery. VAS: visual analog scale. ODI: Oswestry Disability Index.

**Figure 5 diagnostics-13-00229-f005:**
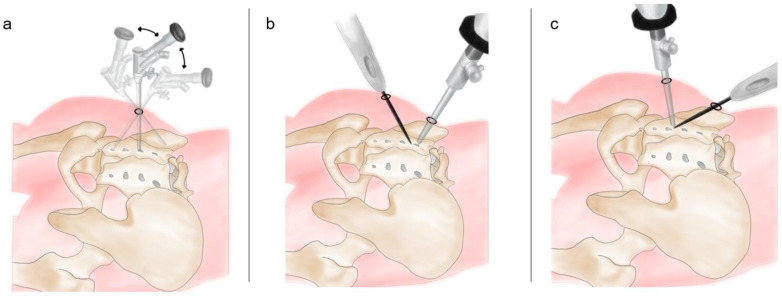
Biportal endoscopy allowed us to improve the working angle by switching the viewing endoscope and the ablation wand. (**a**) Single-port endoscopy has a limited working angle that restricts operating at the S3 lateral branch and L5 dorsal ramus. (**b**,**c**) Biportal endoscopy afforded a more flexible working space by allowing us to exchange the viewing endoscope and the ablation wand.

**Table 1 diagnostics-13-00229-t001:** Summary of the patients and their treatment results.

No	Site	Previous Surgery	PreOP ODI	ODI 1 Month	ODI 6 Months	ODI 12 Months	PreOP VAS	VAS 1 Month	VAS 6 Months	VAS 12 Months
1	Both	RFA *	35	1	5	5	7	1	0	1
2	Both	Fusion, L3-5	25	4	10	10	7	2	1	0
3	Left	Caudal block	40	10	10	10	7	1	2	1
4	Right	Caudal block	28	13	10	10	7	1	2	1
5	Both	Fusion, L4-S1	40	23	16	22	8	1	2	2
6	Right	Epidural block	26	10	10	9	7	1	1	1
7	Both	Epidural block	31	18	18	15	7	1	1	1
8	Left	Fusion, L4-5	30	16	16	16	7	1	2	1
9	Right	Fusion, L4-S1	29	19	9	20	6	1	0	1
10	Both	Fusion, L2-5	37	9	16	10	7	1	1	1
11	Both	Fusion, L3-5	39	23	16	16	8	1	1	1
12	Left	Discectomy, L5/S1	47	22	15	16	8	1	0	0
13	Both	RFA	30	5	5	5	7	1	0	0
14	Both	Caudal block	48	22	22	19	8	1	3	3
15	Both	RFA	35	7	9	9	6	2	1	1
16	Both	RFA	30	1	5	5	6	1	0	1

* RFA: Radiofrequency ablation.

## Data Availability

Patient data are included in the article (Table 1).

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
