# Peer review of "Biportal Endoscopic Radiofrequency Ablation of the Sacroiliac Joint Complex in the Treatment of Chronic Low Back Pain: A Technical Note with 1-Year Follow-Up"

_diagnostics, 2023, doi:10.3390/diagnostics13020229_

Round 1

Reviewer 1 Report

I have a few comments:

-A paired t test was used to compare the study findings between groups, howerer the patient sample was quite small (N=16). Were the data distributions tested for normality (e.g., using the Kolmogorov-Smirnov test) to make sure that the t test was appropriate? Otherwise, it would be preferable to use nonparametric statistics (such as the Wilcoxon rank test) and to report data as median and range.

Does the number X after mean values (i.e., mean +/- X) refer to standard deviation?

-Please briefly explain to what extent the limitations reported at the end of the Discussion section could actually affect the study findings and their applicability in clinical practice.

Reviewer 2 Report

Reviewer Comments

Thank you very much for the opportunity to review the manuscript submission entitled: Biportal Endoscopic Radiofrequency Ablation of the Sacroiliac Joint Complex in the Treatment of Chronic Low Back Pain: A Technical Note With 1-Year Follow-Up. The data is interesting and it has a relevant rationale, however, some limitations and constructive comments are pointed below:

General comments:

·       The article requires a thorough review of the English language.

·       I noticed a similar study which used A bipolar radiofrequency probe was used to lesion the posterior capsule of the SIJ as well as the lateral branches of S1, S2, S3, and the L5 dorsal ramus in multiple locations. Find the reference below

o   Choi WS, Kim JS, Ryu KS, Hur JW, Seong JH, Cho HJ. Endoscopic radiofrequency ablation of the sacroiliac joint complex in the treatment of chronic low back pain: a preliminary study of feasibility and efficacy of a novel technique. BioMed Research International. 2016 Dec 25;2016.

Specific comments

Title and Abstract

·       Mention the study design in the study.

·       Use MeSH terms as keywords.

Introduction

·       Please review the scientific background and rationale for the investigation needs to be emphasized?

Methods

·       All the methods section should be discussed in detail.

·       Describe any efforts to address potential sources of bias.

Discussion

·       Give a cautious overall interpretation of results considering objectives, limitations, multiplicity of analyses, results from similar studies.

Round 2

Reviewer 2 Report

The authors have addressed all the queries satisfactorily. The quality of the manuscript is now improved and can be accepted in the current form.